# Comparative Clinical and Histopathological Study of Oral Leukoplakia in Smokers and Non-Smokers

**DOI:** 10.3390/diagnostics15040502

**Published:** 2025-02-19

**Authors:** Andrei-Eduard Șerban, Ioanina Părlătescu, Elena Milanesi, Iulia Andreea Pelisenco, Maria Dobre, Mariana Costache, Șerban Țovaru, Paula Perlea

**Affiliations:** 1Doctoral School, Carol Davila University of Medicine and Pharmacy, 050474 Bucharest, Romania; andrei-eduard.serban@drd.umfcd.ro; 2Faculty of Dentistry, Carol Davila University of Medicine and Pharmacy, 050474 Bucharest, Romania; serban.tovaru@gmail.com (Ș.Ț.); paula.perlea@umfcd.ro (P.P.); 3Faculty of Medicine, Carol Davila University of Medicine and Pharmacy, 050474 Bucharest, Romania; elena.milanesi@umfcd.ro (E.M.); mariana.costache@umfcd.ro (M.C.); 4Victor Babeș National Institute of Pathology, 050096 Bucharest, Romania; pelisenco.iulia@gmail.com (I.A.P.); maria.dobre@ivb.ro (M.D.)

**Keywords:** oral leukoplakia, oral medicine, smoker, oral potentially malignant disorder

## Abstract

**Background/Objectives:** Oral leukoplakia (OLK) is an oral mucosal lesion classified in the oral potentially malignant disorder group and is associated with an increased risk of malignant transformation (MT). The aim of this study was to compare the clinical and histopathological features of two OLK groups, a group of smokers and a group of non-smokers. **Methods:** In this retrospective study, a cohort of 154 patients with OLK was divided into two groups based on the presence of smoking as a major risk factor. OLK diagnoses were established via clinical and histopathological examination. **Results:** Females were more abundant in the non-smoking group than in the smoking group, where males were more abundant (*p* < 0.001). The average age of the smokers was lower than that of the non-smokers (*p* = 0.003). In the smokers, the buccal mucosa was most frequently affected, while in the non-smokers, the gums and the tongue were primarily involved (*p* = 0.016). In female smokers, involvement of the buccal area and multiple-site involvement were statistically significantly more frequently observed compared to that in female non-smokers (*p* = 0.006). Non-dysplastic lesions were predominant in both groups, with severe dysplasia observed more frequently in the non-smokers than in the smokers. MT was higher in the non-smoker group compared to that in the smoker group. **Conclusions:** OLK in smokers is different from OLK in non-smokers concerning female gender involvement, site location, the number of lesions, and the MT rate.

## 1. Introduction

Oral leukoplakia (OLK) is an oral mucosa lesion that is part of a large subgroup of diseases described by the WHO (World Health Organization) as oral potentially malignant disorders (OPMDs) that carry a high risk of malignant transformation. The OPMD group includes oral leukoplakia, erythroplakia, proliferative verrucous leukoplakia, oral submucous fibrosis, oral lichen planus, actinic cheilitis, palatal lesions in reverse smokers, oral lupus erythematosus, dyskeratosis congenita, oral lichenoid lesions, and oral graft-versus-host disease [1]. Of all the members of the OPMD group, OLK is one of the most common and extensively researched. Over time, OLK has been defined in a variety of ways, and the up-to-date definition is a “white plaque of questionable risk having excluded (other) known diseases or disorders that carry no increased risk for cancer” [2]. The prevalence of this disease varies according to different authors. Zhang et al. described an overall prevalence of 1.39% based on 69 studies with 17,524 participants, a prevalence of 1.82% for Europe in population-based studies, and a prevalence of 4.85% for specific population studies [3], while Mello et al. reported a prevalence of 4.11% for OLK (95% CI = 1.98–6.97) [4]. The most frequent oral mucosa sites affected by OLK are the buccal mucosa, the floor of the mouth, and the tongue, but all areas of the oral mucosa can be involved [5]. The most common clinical classification of OLK divides it into a homogeneous form and a nonhomogeneous form, with verrucous, speckled, and nodular subtypes [6].

The OLK risk factors are well known and consist of smoking, alcohol consumption, and betel nut chewing [5]. The European Commission reported in 2023 that 19% of the adult population of Romania had a smoking habit, the same as the European average, with one out of five adults smoking [7]. EUROSTAT reported that 11.8% of individuals smoke fewer than 20 cigarettes/day, and 3.8% smoke more than 20 cigarettes/day [8]. Regarding alcohol consumption, the European Commission reported that 35% of Romanians consumed alcohol in excessive amounts at least once per month in 2019, a figure above the European average of 19% [7]. Local additional factors such as frictional trauma and the presence of a candida infection may exacerbate the lesions, and their removal from the initial evaluation of OLK lesions has been recommended [1].

The main concern of researchers and clinicians is the malignant transformation (MT) risk of OLK lesions, which varies greatly depending on different factors. The most reliable predictors for MT are an advanced age (50+ years), female gender, tongue location, non-homogenous clinical type, and the presence of epithelial dysplasia [9].

Malignant transformation rates vary per author and meta-analysis, from 6.64% to 9.8% [9,10,11]. Bhattarai et al. reported a 7.39 times higher risk of MT in recurrent OLK compared to the non-recurrent form, and a pooled proportion of recurrence at 22% for different surgical interventions [12]. MT is more likely to occur in patients with moderate or severe epithelial dysplasia than in those with mild or no dysplasia [5].

Treatment selection for OLK is an ongoing discussion in oral medicine research, with some authors debating whether surgical treatment may increase the risk of MT [13]. CO_2_ laser treatment is effective, reliable, and associated with a low recurrence rate [12,14], while non-surgical treatments like photodynamic therapy seem to show great prospects [15,16].

Our research supports the continuous debate in the scientific literature regarding the onset of oral squamous cell carcinoma (OSCC), particularly in the absence of identified risk factors [17,18,19]. The development of OSCC may be related to a number of genetic factors or different epigenetic subtypes and mutations [20,21], or to a compromised immune system [22]. Research on oral leukoplakia in individuals with unknown risk factors may provide insights into OSCC development.

The aim of this study was to assess the clinical and histopathological features and MT risk differences between smoker and non-smoker OLK groups in our cohort from Bucharest, Romania.

## 2. Materials and Methods

The present study is an analytic retrospective cohort study that presents a thorough analysis of oral leukoplakia. The differences between patients divided into two subgroups, smokers and non-smokers, were investigated. Participants in the study were identified from the Oral Medicine Department of the Faculty of Dentistry of the Carol Davila University of Medicine and Pharmacy in Bucharest and our private practice. The patients were consulted between 1996 and 2024. The selection criteria included detailed demographic and clinical data as well as a histopathological confirmation of oral leukoplakia diagnosis. The cases with inconclusive clinical and histopathological information or missing informed consent were excluded from this analysis (exclusion criteria). The dysplasia grades were assessed by the same pathologist (C.M.). The demographic data recorded were gender, age, and smoking status. The clinical form of the OLK lesions was classified, as recommended by van der Waal, 2015 [6], as homogenous or non-homogenous (further categorized as verrucous, speckled, or nodular). We also recorded the number of lesions (single or multiple lesions) and the size of the lesions: smaller than 2 cm^2^, between 2 and 4 cm^2^, and larger than 4 cm^2^. The initial lesion location classification used the WHO topography for oral cancer [23], but it included a large number of sites. Thus, we reduced it to 6 anatomical locations as follows: buccal mucosa, tongue, gums, the floor of the mouth, other sites, and multiple sites. For a more efficient approach, as recommended by Zhang et al. [24], we divided the oral mucosa into high-risk sites, which included the tongue mucosa, the floor of the mouth, the soft palate, and low-risk sites, the remaining oral mucosa. The grading of epithelial dysplasia was carried out in accordance with the WHO standards [25]: no dysplasia, mild dysplasia, moderate dysplasia, and severe dysplasia. The result of the candida mycological examination was recorded, as well as the follow-up period and whether malignant transformation occurred. The treatment selection was individualized for each patient, local irritative factors were removed, and smoking reduction or cessation was recommended. The histopathological result was the deciding factor for subsequent treatment, which ranged from clinical and histopathological monitoring to surgical excision depending on the lesion site, dimension, and patient’s general health state.

The ethical approval and the acceptance of the study protocol were obtained from the Ethics Committee of Carol Davila University of Medicine and Pharmacy Bucharest Romania cod PO-35-F-03 number 1139/13.01.2023. This study was performed according to the recommendations of the Declaration of Helsinki. Written informed consent was obtained from all the patients.

Statistical Package for the Social Sciences (SPSS version 17.0) was used to conduct the statistical analysis. Differences in age between the two groups were assessed using the *t*-test, while for categorical variables, statistical significance was evaluated using the chi-square test or Fisher’s exact test.

## 3. Results

We analyzed the clinical and histopathological data of 154 OLK patients to better comprehend the distinctions attributable to smoking in the present study. The cohort included 88 females (57.1%) and 66 males (42.9%) with an overall mean age of 55.16 ± 12.45 (min 26–max 86 years). The cohort was divided into two groups: 108 smokers (including 23 ex-smokers at the time of diagnosis—patients who had quit smoking for at least 1 year) and 46 non-smokers. Table 1 presents data on the demographic and clinical features of OLK lesions in both groups of patients. The average age of the smokers was significantly lower than that of the non-smokers (53.22 years vs. 59.67 years) (*p* = 0.003). Of the 108 smokers, *n* = 58 (53.70%) were male and *n* = 50 (46.30%) were female. In the non-smoker group, there were more females (*n* = 38; 82.61%) compared to males (*n* = 8; 17.39) (*p* < 0.001, χ^2^ = 17.369). We detected that single lesions were more frequent in the non-smoker group (76.09%) compared to (51.85%) the smokers (*p* = 0.005, χ^2^ = 7.838).

Comparing the lesion sites of smokers vs. non-smokers, we found an overall significant difference (*p* = 0.016): in smokers, the buccal mucosa was more frequently involved (28.70%) than in non-smokers (15.22%), along with the floor of the mouth (7.41% vs. 2.17%). In the non-smoker group, the tongue and the gums were more frequently involved compared to the smoker group (for the tongue 21.74% vs. 6.48%, and for gums 30.43% vs. 26.85%). No statistical difference was found regarding the high-risk and low-risk sites. In the smoker group, the high-risk sites were more involved compared to the non-smoker group (35.29% vs. 28.26%). Meanwhile, in the non-smoker group, the low-risk sites were more involved (71.74% vs. 64.81%).

OLK clinical forms showed no significant differences between the two groups, with a similar frequency of homogenous forms in the non-smokers (69.57%) and in the smokers 62.04%. The non-homogenous form was more frequently encountered in smokers (37.96%) than in the non-smoker group (30.43%), without a significant difference.

Comparing the two groups, there was no significant difference in terms of lesion size, with the most lesions being under 2 cm^2^ in both groups, followed by 2–4 cm^2^ and larger than 4 cm^2^.

Candida examination was performed in 70 patients (53 smokers and 17 non-smokers) and was found to be positive in 50.94% of smokers and 41.18% of non-smokers.

Two separate analyses stratifying the cohort by sex were conducted to assess whether there was a significant association between the smoker status and the following variables: clinical form, site of the OLK lesion, number of lesions, location by risk, size, and presence of candida infection. When considering the female group (*n* = 88), a significant association between the site of the OLK lesion and smoker status was found (*p* = 0.006): the buccal area was affected in 28% of the smokers, while in the non-smokers, the buccal area was affected only in 15.22% of the cases. Moreover, multiple site lesions were found in 30% of female smokers and in 21.5% of non-smokers. Another significant association between female smokers and non-smokers was found considering the number of lesions (*p* = 0.02). The same analysis was conducted on the male group, and no significant associations were found for all the tested variables considering smoker status (Table 2).

No significant associations in the female group were found for the other variables considering smoker status.

### Histopathological Results

The histopathological data are reported in Table 3. The analysis revealed a similar frequency of the non-dysplastic lesions in both groups (56.48% smokers vs. 60.87% in non-smokers). Mild dysplasia was found in 37.04% of patients in the smoker group and 26.09% of patients in the non-smoker group. Severe dysplasia was found more frequently in non-smokers (8.70% vs. 3.70% in smokers), without reaching statistical significance.

The mean period of follow-up for the cohort was 35.98 (1–228) months, 36.40 months for the smokers and 35.05 months for non-smokers.

During follow-up, malignant transformation occurred in 15 patients (12.71%), 8 smokers (8.99%), and 7 non-smokers (18.92%). Although this event was more frequent in the non-smokers, statistical significance was not reached. The MT rate per year for the entire cohort was 4.23%, while that in the smoker group was 2.63% and that in the non-smoker group was 2.39%.

## 4. Discussion

Since the World Health Organization’s initial definition of OLK in 1978 [26], smoking has been acknowledged as a significant factor in the etiology of OLK; however, there is a scarcity of research evaluating the clinical and histopathological characteristics that differentiate OLK according to smoking status in the literature [27,28,29,30,31,32,33,34]. Thus, we investigated the clinical and histopathological differences in OLK in smokers and non-smokers. There is a large variation regarding the clinical features of OLK, based mainly on geographic distribution and in close connection with population habits.

Our cohort included 154 cases of oral leukoplakia, divided into two groups as follows: 108 cases of OLK in smokers (70.12%) and 46 cases of OLK in non-smoker patients (29.88%). The present findings reveal that OLK was diagnosed more frequently in women than in males (57.10% vs. 42.90%) in the whole cohort. This result is consistent with reports from Holland [27] that found 140 OLK cases, 69% in women, and a study from Beijing [28] on 875 OLK cases that reported 57.3% to be female patients. Conversely, Kusiak et al. [29] reported a study of 416 OLK cases where 52.9% of patients were male. A population-based cohort study evaluating 1888 histopathologically confirmed OLK cases in Northern California found that 57.36% of patients were male [30], and a single-center study of 676 cases in Japan [31] reported more men than females (53.7% vs. 46.3%).

We observed that the non-smoking group had a predominance of females (82.61%), and there was a higher proportion of men (53.39%) in the smoking group (*p* < 0.001, χ^2^ = 17.369). This is comparable to a previous study conducted in Spain [32] on 52 OLK cases, which found that 82% of non-smokers were female, and 78% of smokers were male. This result is consistent with a 2012 Brazilian study [33] that found that the smoker group was dominated by men, and the non-smoker group was dominated by women. On the other hand, Schepman et al. [34] report that women are predominant in both in the non-smoker cohort and the entire cohort.

The mean age was higher in the non-smoker group than in the smoker group, similar to Freitas et al. [32]. This is similar to the mean age described in OLK in a study by Kokubun et al., where in 676 OLK cases, the onset age was >50 years [31]. However, as stated by the researchers, the data sheets of the patients in the Japanese cohort did not include information regarding smoking habits or alcohol intake.

The homogeneous clinical form of OLK was most frequently encountered in all patients, unrelated to smoking habits—62.04% in smokers, and 69.57% in non-smokers. This result is consistent with the reports of Evren et al. [27], which found no correlation between tobacco usage and the clinical form of OLK, and is different from previous reports that found a strong relationship between homogenous OLK and smoking [29] or detected a higher frequency in smokers. In our observations, in the smoker group, the most frequently affected site was the buccal mucosa followed by gums and multiple sites, and in non-smokers the gums were followed by multiple sites. The majority of OLK lesions were less than 2 cm^2^ in size. Kokubun et al. [31] found the tongue and the gums to be the primary sites of involvement in their OLK cohort, differentiating them only by sex and not by other criteria.

Dysplasia was identified in 65 cases (42.20%) in all OLK patients. The percentages of smokers and non-smokers were slightly different (43.52% vs. 39.13%). This outcome is different from other studies: Chaturvedi et al. [30] reported 15% dysplasia, Pentenero et al. [34] reported 15.8%, and Evren et al. [27] reported 60%. Consistent with the findings of Evren et al. [27], in our study, smoking did not correlate with the histopathological categories.

Regarding the MT rate, in 15 of our cohort cases, MT occurred. The percentage was 12.71% for the entire cohort, with an average of 4.23% per year. In a previous study [35] on a smaller series of OLK patients, we reported an MT rate of 7.5%. The outcome of the present study is higher than average, as Aguire-Urizar et al. [9] report an OLK MT rate of 9.8% for 5 years (2015–2020) (95% CI: 7.9–11.7), and Pimenta-Barros et al. [11] found a pooled MT rate of 6.64% (95% CI: 5.21–8.21) in a meta-analysis with 55 studies including 41,231 OLK cases published before June 2024. Guan et al. [10] reported an MT pooled MT rate of 7.20% (95% CI: 5.40–9.10) for 26 OLK studies published in the last 20 years (2000–2022). We also observed that the non-smokers’ group had almost twice the MT rate compared to the smokers’ group, with 9.88% for the smokers (8 out of 108) and 18.92% for the non-smoker group (7 out of 46). Also, we observed that all seven cases from the non-smokers’ group with MT were in female subjects. The predominance of malignant transformation in females is well established, even though to our knowledge no previous research has provided the rate of malignant transformation for distinct non-smoking OLK groups [5,10,34]. This is in agreement with Pentenero et al. [36], who report that DNA ploidy might identify a subset OLK with higher epithelial dysplasia manifestation and higher malignant transformation risk, even though its value in predicting MT is limited and further studies are needed. Treatment for oral leukoplakia is an ongoing discussion; in a systematic review, Lodi et al. [37] found that there is no evidence of a treatment that is effective in preventing the development of oral cancer, with some being effective in healing oral lesions but not preventing relapse and side effects. Oral malignancy remains one of the most aggressive cancers in humans, with mortality rates of 13.6% reported for Europe in 2020 [38].

The current study has some limitations. The study group size was constrained and contingent upon patient accessibility, needs, and compliance with follow-up appointments. The methodology used was not identical to that used in previously published research. We did not consider alcohol consumption, as it is challenging to accurately estimate the specific amount of alcohol intake in current clinical practice. In the Introduction, we detailed alcohol intake as reported by the EUROSTAT in the general population, considering it a major risk factor for malignant transformation. A further disadvantage, mostly attributable to the study’s retrospective design, was the absence of current data concerning emerging smoking modalities (such as vaping, e-cigarettes, and nicotine pouches) and other novel potential risk factors. Therefore, we emphasize the need for new prospective research regarding the impact of new risk factors on oral mucosal lesions.

The strengths and novelty of this research lie in our aim to differentiate and compare oral leukoplakia lesions in patients exposed to smoking as a primary risk factor. We found few studies that reported oral leukoplakia from this point of view, and those we found used other methodologies and different types of analysis.

As we observed that patients with unknown risk factors usually progress with a worse clinical outcome, we consider that there is a need for genetic testing and immunohistochemistry tests to select the most appropriate individualized treatment approach. We recommend histopathological examination in all oral leukoplakia lesions, as it remains the most reliable method for evaluating the malignant transformation potential. Subsequently, biopsies should be performed during follow-up, along with other tests in case of clinical changes in the lesions or the onset of clinical symptoms.

## 5. Conclusions

Our study revealed statistical differences between OLK in smokers and non-smokers concerning gender—with a higher proportion of females in the non-smoker group. The most commonly affected areas were smokers’ buccal mucosa and non-smokers’ gums. The homogeneous type was the most encountered clinical form of OLK in all patients and was unrelated to smoking behavior. The presence of dysplasia did not significantly differ between smokers and non-smokers.

We report that the non-smokers had an almost twofold higher rate of malignant transformation than the smokers, and that all of the non-smokers’ malignant transformation cases were in women.

## Figures and Tables

**Table 1 diagnostics-15-00502-t001:** Demographic and clinical characteristics: comparison between smokers and non-smokers.

Variable	Smokers (*n* = 108)	Non-Smokers (*n* = 46)	*p*-Value
**Age**	53.22 ± 11.27	59.67 ± 13.96	** *0.003* **
**Gender**			
*Male*	58 (53.70%)	8 (17.39%)	***<0.001*** χ^2^ = 17.369
*Female*	50 (46.30%)	38 (82.61%)	
**Clinical form**			
*Homogeneous*	67 (62.04%)	32 (69.57%)	0.372 χ^2^ = 0.79
*Non-homogeneous*	41 (37.96%)	14 (30.43%)	
**Site of OLK lesion**			
*Buccal*	31 (28.70%)	7 (15.22%)	***0.016*** (Fisher Test)
*Tongue*	7 (6.48%)	10 (21.74%)	
*Gums*	29 (26.85%)	14 (30.43%)	
*Floor of mouth*	8 (7.41%)	1 (2.17%)	
*Other sites*	4 (3.71%)	5 (10.87%)	
*Multiple sites*	29 (26.85%)	9 (19.57%)	
**Number of lesions**			
*Single*	56 (51.85%)	35 (76.09%)	***0.005*** χ^2^ = 7.838
*Multiple*	52 (48.15%)	11 (23.91%)	
**Location by risk**			
*High risk*	38 (35.29%)	13 (28.26%)	0.403 χ^2^ = 0.698
*Low risk*	70 (64.81%)	33 (71.74%)	
**Size**			
*<2* cm^2^	55 (50.93%)	21 (45.65%)	0.278 χ^2^ = 2.562
*2–4* cm^2^	29 (26.85%)	18 (39.13%)	
*>4* cm^2^	24 (22.22%)	7 (15.22%)	
**Candida infection ***			
*Negative*	26 (49.06%)	10 (58.82%)	0.483 χ^2^ = 0.492
*Positive*	27 (50.94%)	7 (41.18%)	

* Available for 53 smokers and 17 non-smokers.

**Table 2 diagnostics-15-00502-t002:** Clinical characteristics by gender groups: comparison between smokers and non-smokers.

Female Group (*n* = 88)			
Variable	Smokers (*n* = 50)	Non-Smokers (*n* = 38)	*p*-Value
**Site of OLK lesion**			
*Buccal*	14 (28.00%)	4 (10.53%)	***0.006*** (Fisher Test)
*Tongue*	3 (6.00%)	10 (26.32%)	
*Gums*	15 (30.00%)	11 (28.94%)	
*Floor of mouth*	3 (6.00%)	1 (2.63%)	
*Other sites*	0 (0%)	4 (10.53%)	
*Multiple sites*	15 (30.00%)	8 (21.05%)	
**Number of lesions**			
*Single*	26 (52%)	29 (76.32%)	***0.02*** χ^2^ = 5.447
*Multiple*	24 (48%)	9 (23.68%)	
**Male group (*n* = 88)**			
**Variable**	Smokers (*n* = 58)	Non-Smokers (*n* = 8)	*p*-value
**Site of OLK lesion**			
** *Buccal* **	17 (29.31%)	3 (37.50%)	0.856 (Fisher Test)
** *Tongue* **	4 (6.90%)	0 (0%)	
** *Gums* **	14 (24.13%)	3 (37.50%)	
** *Floor of mouth* **	5 (8.63%)	0 (0%)	
** *Other sites* **	4 (6.90%)	1 (12.50%)	
** *Multiple sites* **	14 (24.13%)	1 (12.50%)	
**Number of lesions**			
** *Single* **	30 (51.72%)	6 (75.00%)	0.275 (Fisher Test)
** *Multiple* **	28 (48.28%)	2 (25.00%)	

**Table 3 diagnostics-15-00502-t003:** Histopathological evaluation, follow-up, and outcome.

Variable	Smokers (*n* = 108)	Non-Smokers (*n* = 46)	*p*-Value
**Histopathological evaluation**			
*No dysplasia*	61 (56.48%)	28 (60.87%)	0.071 (Fisher Test)
*Mild dysplasia*	40 (37.04%)	12 (26.09%)	
** *Moderate dysplasia* **	4 (3.70%)	4 (8.70%)	
** *Severe dysplasia* **	3 (2.78%)	2 (4.34%)	
**Presence of dysplasia**			
** *No dysplasia* **	61 (56.48%)	28 (60.87%)	0.255 χ^2^ = 0.492
** *Dysplasia* **	47 (43.52%)	18 (39.13%)	
**Mean period of follow-up ***	36.40 months	35.05 months	
**Outcome ****			
** *Malignant transformation* **	8 (9.88%)	7 (18.92%)	0.171 χ^2^ = 1.872
** *Non-malignancy* **	73 (90.12%)	30 (81.08%)	

* Available for 86 smokers and 39 non-smokers; ** available for 81 smokers and 37 non-smokers.

## Data Availability

The data presented in this study are available on reasonable request from the corresponding author.

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
