# Peer review of "Comparative Clinical and Histopathological Study of Oral Leukoplakia in Smokers and Non-Smokers"

_diagnostics, 2025, doi:10.3390/diagnostics15040502_

Round 1

Reviewer 1 Report

Comments and Suggestions for Authors

Rather small group of patients (154), and followed for a rather short period of time (3years; 36 months), dysplasia in 52 mild or low grade and 13 with moderate and severe or high grade, with the result of high number of carcinoma (15!, almost 10%).

Does it mean that everyone from high grade dysplasia progress into carcinoma!?

What about applied treatment?

I think it should be explained more than just smoking. The only difference among smoker/nonsmoker group is at level of low grade dysplasia, on carcinoma is frequent in nonsmoker group (8 (9.88%) vs 7 (18.92%)).

In another paper you could elaborate the treatment options for those with low grade and those for high grade dysplasia, because in mine opinion the number of malignancy is very high (15 carcinoma from 65 dysplasias; 13 high grade)

Author Response

For research article “Comparative clinical and histopathological study of oral leukoplakia in smokers and non-smokers

Response to Reviewer 1 Comments

1. Summary

Thank you very much for taking the time to review this manuscript! Your comments improved it. Please find the detailed responses below and the corresponding revisions in track changes in the re-submitted file.

Our best regards!

2. Questions for General Evaluation

Yes

Can be improved

Must be improved

Not applicable

Does the introduction provide sufficient background and include all relevant references?

(x)

( )

( )

( )

Is the research design appropriate?

(x)

( )

( )

( )

Are the methods adequately described?

(x)

( )

( )

( )

Are the results clearly presented?

(x)

( )

( )

( )

Are the conclusions supported by the results?

(x)

( )

( )

( )

Response and Revisions

Thank you for your effort!  We did all the suggested changes in the point- by -point response added below.

Comments 1: Rather small group of patients (154), and followed for a rather short period of time (3years; 36 months), dysplasia in 52 mild or low grade and 13 with moderate and severe or high grade, with the result of high number of carcinoma (15!, almost 10%).

Response 1:  Thank you for pointing this out. We agree with this comment, but the number of patients is not controllable in any way by us, and is dependent on the addressability of the Oral Medicine clinic and the patient needs. We  mentioned this aspect in the Discussion Chapter and updated lines 271-273.

 The study group size is constrained and contingent upon patient accessibility, needs and their compliance with follow-up appointments.

Comments 2: Does it mean that everyone from high grade dysplasia progress into carcinoma!?

Response 2: Thank you for this observation! It is right that not all high grade dysplasia cases for sure progress to carcinoma, but it is acceptable by all researchers that it carries the highest risk to progress to carcinoma in cases left untreated.

We added in the manuscript the following phrase: Lines 65-66: MT is more likely to occur in patients with moderate or severe epithelial dysplasia than in those with mild or no dysplasia[5].

 Comments 3: What about applied treatment?

 Response 3:  Thank you for this question. Treatment was individualized in each patient following the van der Waal recommendations from 2015. We did not analyze the treatment options in this study,as it was not the aim of this research. We will address this matter in another study. The treatment options were detailed in Material and Methods Chapter, lines 105-106: “The treatment selection was individualized for each patient using the recommended protocol described by Van de Waal, 2015 [6]

Comments 4: I think it should be explained more than just smoking. The only difference among smoker/nonsmoker group is at level of low grade dysplasia, on carcinoma is frequent in nonsmoker group (8 (9.88%) vs 7 (18.92%))

Response 4: Thank you for pointing this out. This research aimed to analyze the impact of smoking on oral leukoplakia lesions. The statistically relevant differences were in smoking and no dysplasia. We also observed a higher frequency of mild dysplasia in smokers 37.04% vs 26.09% in non-smokers and carcinoma is frequent in the nonsmoker group (8 (9.88%) vs 7 (18.92%)).

We updated the Discussion Chapter as follows: As we observed that patients with unknown risk factors usually progress with a worse clinical outcome, we consider that there is a need for genetic testing and immunohistochemistry tests to select the most appropriate individualized treatment approach.” lines 287-290

Comments 5: In another paper you could elaborate the treatment options for those with low grade and those for high grade dysplasia, because in mine opinion the number of malignancy is very high (15 carcinoma from 65 dysplasias; 13 high grade)

 Response 5: Thank you for your insightful suggestion! We intend to do this kind of research, but we will enlarge the cohort to correlate treatment and results.

Reviewer 2 Report

Comments and Suggestions for Authors

Dear Authors/Editors,

Thank you for the opportunity to review the manuscript submitted to DiagnosticsComparative clinical and histopathological study of oral leukoplakia in smokers and non-smokers.

After analysis, we present the following considerations:

Although the evaluated article does not introduce innovations of global scope, it demonstrates considerable relevance in the regional context, contributing to the better understanding and practical application of issues specific to its field of study.

It is worth noting that the statistical analysis conducted by the authors was not subjected to a detailed technical evaluation by the reviewers, as this is not the primary area of expertise of the review team.

Nonetheless, the article is well-structured, adheres to academic standards, and is deemed suitable for publication.

Author Response

For research article “Comparative clinical and histopathological study of oral leukoplakia in smokers and non-smokers

Response to Reviewer 2 Comments

1. Summary

Thank you very much for taking the time to review and appreciate our manuscript! We improved the manuscript according to the other reviewers' recommendations. Please find the corresponding revisions in track changes in the re-submitted file.

Our best regards!

  1. Questions for General Evaluation

Yes

Can be improved

Must be improved

Not applicable

Does the introduction provide sufficient background and include all relevant references?

(x)

( )

( )

( )

Is the research design appropriate?

(x)

( )

( )

( )

Are the methods adequately described?

(x)

( )

( )

( )

Are the results clearly presented?

(x)

( )

( )

( )

Are the conclusions supported by the results?

(x)

( )

( )

( )

Reviewer 3 Report

Comments and Suggestions for Authors

Dear Authors,

I appreciate the research you conducted, as oral leukoplakia is considered a potentially malignant disorder. However, there are a few observations regarding this manuscript, which I consider important to be improved:

In the Abstract, the Material and Method is very short while Results are too long, you present a lot of unnecessary data, difficult to follow. On the other hand, you could pay more attention to the Conclusions, which are schematic and somehow unclear.

In the Introduction you discuss about definition, prevalence, risk factors, malignant transformation. In my opinion, lines 58- 66 would better serve in Discussions. I would expect you to say more about the oral squamous cell carcinoma as despite recent developments of treatment strategies, this malignancy still remains one of the most aggressive with high mortality rates. This section needs improvements, try to offer a more coherent transition from one paragraph to another, otherwise it looks like information gathered in a very schematic way.

In Materials and Methods I found no referral to what type of study you conducted, only at the end of the manuscript you define it as a retrospective one. You evaluate patients who presented in your clinics between 1996-2024 and the criteria you use were based on references from 2001, 2015, 2024. Please explain how you applied these to the early cases. The mean follow-up periods were around 35 months, please mention the variation limits and define more clearly inclusion and exclusion criteria.  

The Results contain no information regarding the male patients. Table 3 is not included in the text, many percentages are inserted and they are not clearly explained.  The most important part is the malignant transformation (lines 169-173) and you declare that there ``was no statistically significant difference between both groups``. Please specify what histological types showed malignant transformation. What was the gender distribution of malignant transformation in the smokers group? In my opinion, malignant transformation is the most important aspect and I consider you should add more information to this section.

Regarding Conclusions, I would like to ask you why you consider conclusion an aspect that showed no statistical significant difference between groups : Lines 274-276``We report an almost double malignant transformation rate for the non-smokers group compared to smokers group, with all of the malignant transformation for the non smokers group occurring in women.``

Comments on the Quality of English Language

Please ask support for English language correction (please revise spelling, punctuation and grammar). 

Author Response

For review article “Comparative clinical and histopathological study of oral leukoplakia in smokers and non-smokers

Response to Reviewer 3 Comments

1. Summary

Thank you very much for taking the time to review this manuscript! Your recommendations improved the article. Please find the detailed responses below and the corresponding revisions/corrections highlighted/in track changes in the re-submitted file.

2. Questions for General Evaluation

Reviewer’s Evaluation

Response and Revisions

Yes

Can be improved

Must be improved

Not applicable

Thank you for your effort!  We did all the suggested changes in the point- by -point response added below.

Does the introduction provide sufficient background and include all relevant references?

( )

( )

(x)

( )

Is the research design appropriate?

( )

( )

(x)

( )

Are the methods adequately described?

( )

( )

(x)

( )

Are the results clearly presented?

( )

( )

(x)

( )

Are the conclusions supported by the results?

( )

( )

(x)

( )

Comments 1: In the Abstract, the Material and Method is very short while Results are too long, you present a lot of unnecessary data, difficult to follow. On the other hand, you could pay more attention to the Conclusions, which are schematic and somehow unclear.

Response 1: Thank you for your constructive comment! Therefore, we removed the following phrase from the results “Single lesions were more frequent in the non-smokers group compared to smokers (p=0.005).”  We also made the changes in Material and Method and Conclusions. The updated text in the manuscript is the following:

Lines 15-18: “Materials and methods: In this retrospective study a cohort of 154 OLK patients  were divided into two groups, based on the presence of smoking as a major risk factor. The OLK diagnosis was established by clinical and histopathological examination. “

Lines 26-27: “Conclusion: OLK in smokers is different from OLK in non-smokers concerning the female gender involvement, site location, number of lesions, MT rate.”

Comments 2: In the Introduction you discuss about definition, prevalence, risk factors, malignant transformation. In my opinion, lines 58- 66 would better serve in Discussions. I would expect you to say more about the oral squamous cell carcinoma as despite recent developments of treatment strategies, this malignancy still remains one of the most aggressive with high mortality rates. This section needs improvements, try to offer a more coherent transition from one paragraph to another, otherwise it looks like information gathered in a very schematic way.

Response 2:  Thank you for this recommendation. We have accordingly revisited the manuscript.

In the Introduction, in lines 62-63  we added the following phrase: Malignant transformation rates vary between different authors and  meta-analyses from 6.64% to 9.8% [9-11]”.

We revisited the Discussion chapter and included the data previously mentioned in the Introduction. The updated text is the following: LINES 248-259 Regarding the MT rate, in 15 of our cohort cases MT occurred, that is 12.71% for the entire cohort with an average of 4.23% per year.  In a previous study [36] on a smaller series of OLK patients we reported a MT rate of 7.5%. The outcome of the present study is higher than the average, as Aguire-Urizar et al [9] report OLK a  9.8% MT rate for 5 years (2015-2020) (95% CI: 7.9-11.7),  and Pimenta-Barros et al [11] found a pooled MT rate of 6.64% (95% CI: 5.21-8.21) in a meta-analysis with 55 studies including  41231 OLK cases published before June 2024. Guan et al [10] report a MT pooled MT rate of 7.20% (95% CI: 5.40-9.10) for 26 OLK studies published in the last 20 years(2000-2022). We also observed that the non-smokers' group had almost twice the value of MT rate compared to the smokers' group, 9.88% for the smokers (8 out of 108) and for the non-smokers group 18.92%(7 out of 46). Also, we observed that all 7 cases from the non-smokers' group with MT were in female subjects.

 Comments 3: In Materials and Methods I found no reference to what type of study you conducted, only at the end of the manuscript you define it as a retrospective one. You evaluate patients who presented in your clinics between 1996-2024 and the criteria you use were based on references from 2001, 2015, 2024. Please explain how you applied these to the early cases. The mean follow-up periods were around 35 months, please mention the variation limits and define more clearly inclusion and exclusion criteria.

Response 3: Thank you for pointing this out!

 We added the phrase: The present study is an analytic retrospective cohort study that aims to present a thorough analysis of oral leukoplakia patients investigating the differences between OLK cases divided into two subgroups, smokers and non-smokers” in lines 82-84

For the present research we included from our database, only the cases that were retrospectively analyzed and followed all the new criteria.

Inclusion and exclusion criteria were revisited accordingly in lines 87-91. “The selection criteria included detailed demographic and clinical data as well as histopathological confirmation for oral leukoplakia diagnosis. The cases with inconclusive clinical and histopathological information or missing informed consent were excluded from this analysis (exclusion criteria). ”

We added variation limits from (1-228 months) in lines 184-185.

Comments 4: The Results contain no information regarding the male patients.

Response 4: The same analysis performed in the female group was performed in the male group. No significant association between the smoker status and the clinical variables was found. The results have been added in the text (lines 174-175):The same analysis was conducted on the male group, and no significant associations were found for all the tested variables considering the smoker status.”

Comments 5: Table 3 is not included in the text,

Response 5: Thank you for pointing this out and do excuse our mistake! Table 3 is now mentioned in line 194.

Comments 6: many percentages are inserted and they are not clearly explained.  The most important part is the malignant transformation (lines 169-173) and you declare that there ``was no statistically significant difference between both groups``. 

Response 6: The malignant transformation was more frequent in the nonsmokers group, however without reaching the statistical significance of p<0.05. This could be due to the relatively small sample size. The sentence “The was no statistically significant difference between both groups` has been correctly rephrased: lines 187-188 “Although this event was more frequent in non-smokers, the statistical significance was not reached”

Comments 7: Please specify what histological types showed malignant transformation. What was the gender distribution of malignant transformation in the smokers group? In my opinion, malignant transformation is the most important aspect and I consider you should add more information to this section.

Response 7: Only 8 smokers developed malignant transformation (5 female and 3 males). Your suggestion is very interesting, however, the limited number of patients developing malignant transformation does not allow us to perform additional statistical analysis by sex.

Comments 8: Regarding Conclusions, I would like to ask you why you consider conclusion an aspect that showed no statistical significant difference between groups : Lines 274-276``We report an almost double malignant transformation rate for the non-smokers group compared to smokers group, with all of the malignant transformation for the non smokers group occurring in women.``

Response 8:  Thank you for your observation! Despite not being a statistically relevant outcome between the two groups, in the context of the current scientific literature, we find it worth mentioning in the conclusions. Its importance is related to clinical applicability for the patients diagnosed with these lesions, mostly for non-smoker women, for whom the clinical implications may be worse.

Reviewer 4 Report

Comments and Suggestions for Authors

Dear Authors

Thank you for the submission of your manuscript. The manuscript offers a comparative clinical and histopathological study of oral leukoplakia (OLK) in smokers and non-smokers, aiming to explore differences in their characteristics and the associated risk of malignant transformation.

Please find my comments-

1.  Please rephrase “Oral leukoplakia (OLK) is a lesion of the oral mucosa part of the Oral Potentially Malignant Disorders group, a group of lesions that carry increased risk of malignant transformation” in the Abstract as “Oral leukoplakia (OLK) is an oral mucosal lesion classified under Oral Potentially Malignant Disorder group, associated with an increased risk of malignant transformation”.

2. In the Introduction, briefly describe Oral Potentially Malignant Disorders and enumerate them. Thereafter, describe Oral lekoplakia and mention OLK is the most common OPMD.

3. Other known risk factors for oral leukoplakia also needs to be included.

4. Mention the study design in Materials and methods section.

5. The study does not address newer smoking methods (e.g., vaping, e-cigarettes) which could affect the outcomes and relevance of the findings.

6. How was the sample size estimation done?

7. The manuscript raises serious concerns on the English language. The manuscript needs to be run on Grammarly for grammar and spell check.

Comments on the Quality of English Language

The manuscript raises serious concerns on the English language. The manuscript needs to be run on Grammarly for grammar and spell check.

Author Response

For review article “Comparative clinical and histopathological study of oral leukoplakia in smokers and non-smokers”

Response to Reviewer 4 Comments

  1. Summary

Thank you very much for taking the time to review this manuscript! Please find the detailed responses below and the corresponding revision in track changes in the re-submitted file.

  1. Questions for General Evaluation

Yes

Can be improved

Must be improved

Not applicable

Does the introduction provide sufficient background and include all relevant references?

( )

(x)

( )

( )

Is the research design appropriate?

( )

(x)

( )

( )

Are the methods adequately described?

( )

( )

(x)

( )

Are the results clearly presented?

( )

(x)

( )

( )

Are the conclusions supported by the results?

( )

(x)

( )

( )

Response and Revisions

Thank you for your effort!  We did all the suggested changes in the point- by -point response added below.

Comments 1:  Please rephrase “Oral leukoplakia (OLK) is a lesion of the oral mucosa part of the Oral Potentially Malignant Disorders group, a group of lesions that carry increased risk of malignant transformation” in the Abstract as “Oral leukoplakia (OLK) is an oral mucosal lesion classified under Oral Potentially Malignant Disorder group, associated with an increased risk of malignant transformation”.

Response 1: Thank you for your suggestion! We did the changes in lines 12-13.

 Oral leukoplakia (OLK) is an oral mucosal lesion classified under Oral Potentially Malignant Disorder group, associated with an increased risk of malignant transformation (MT)

Comments 2: In the Introduction, briefly describe Oral Potentially Malignant Disorders and enumerate them. Thereafter, describe Oral lekoplakia and mention OLK is the most common OPMD.

Response 2: Thank you for your suggestion. We did the changes in lines 32-37:The OPMD group includes oral leukoplakia, erythroplakia, proliferative verrucous leukoplakia, oral submucous fibrosis, oral lichen planus, actinic cheilitis, palatal lesions in reverse smokers, oral lupus erythematosus, dyskeratosis congenita, oral lichenoid lesions, oral graft versus host disease [1]. Of the OPMD group, OLK is one of the most frequent and extensively researched.”

Comments 3: Other known risk factors for oral leukoplakia also needs to be included.

Response 3: Thank you for your recommendation! We added other factors in lines 55-57

The local additional factors such as frictional trauma and the presence of Candida infection may exacerbate the lesions and it is recommended their removal from the initial consultation [1].”

Comments 4:  Mention the study design in Materials and methods section.

Response 4: Thank you for pointing this out! We added lines 82-84

The present study is an analytic retrospective cohort study that aims to present a thorough analysis of oral leukoplakia patients investigating the differences between OLK cases divided into two subgroups, smokers and non-smokers.

Comments 5: The study does not address newer smoking methods (e.g., vaping, e-cigarettes) which could affect the outcomes and relevance of the findings.

Response 5: Thank you for this comment. We address this matter, in the discussion section, lines 277-282 “A further disadvantage, mostly attributable to the study's retrospective design, is the absence of current data concerning emerging smoking modalities (such as vaping, e-cigarettes, nicotine pouches, etc.) and other novel potential risk factors.  Therefore, we emphasize the need for new prospective research regarding the impact of new risk factors on oral mucosal lesions.”

Comments 6: How was the sample size estimation done?

Response 6: Thank you for the comment. The size estimation was not done. The present research is a retrospective analysis of the OLK database of the patients. The patients are referred mainly by dentists, general practitioners, and ENT doctors. The population size cannot be defined. We added in the manuscript lines 271-273.

“The study group size is constrained and contingent upon patient accessibility, needs and their compliance with follow-up appointments.”

Comments 7: The manuscript raises serious concerns on the English language. The manuscript needs to be run on Grammarly for grammar and spell check.

Response 7: Thank you for your recommendation! We performed a meticulous grammar and spell check of the whole manuscript.

Round 2

Reviewer 3 Report

Comments and Suggestions for Authors

Dear Authors,

Although you made some improvements to your paper, I still consider that some important aspects have not been properly revised.

Introduction

Lines 71-76 are difficult to follow and do not seem to have any connection o the purpose of your study. Moreover, since you conduct a study regarding prevalence and other characteristics of OLK (gender distribution, histological subtypes etc.), the appropriate treatment strategy is not the object of your research.

Material and Methods:

In this section, it is mentioned that this was a retrospective cohort study. If so, how could the patients offer their informed consent? For these types of study, an ethics committee generally gives the approval for the reasonable use of available clinical and histological data.  Again, the treatment protocol is not relevant for your research and since it is guided by a reference published in 2015, how could it be applied to cases identified before 2015?

Results:

This section is still difficult to follow, you mention many percentages and results of Chi-square-tests in the text, which is actually appropriate for a table. I still consider that Table 2 should have included also the descriptive data for the male patients. Just because they did not reach statistical significance, it does not mean they are not worth being mentioned. Apart from that, it should be kept in mind, that on the other side, the pure achievement of statistical significance does not automatically imply a relevant clinical significance.

Since the follow- up time of the patients was so variable, ranging from 1 to 228 months, it is difficult to make clear assumptions regarding malignant transformation, as patients who were longer followed have higher odds of being at one point diagnosed with a malignant transformation. Moreover, you included the ex-smokers in the smokers group, which I find reasonable, however, it still matters how long ago these quit smoking.  

Discussion and Conclusions:

In spite of the previously mentioned concerns, the Conclusion section has not been improved. It should have been clearly mentioned, that your conclusion is more a clinical observation, maybe randomly encountered, and not a statistically and clinically validated statement.

Comments on the Quality of English Language

The English language has not been improved, the topic of phrases is still difficult to follow. 

Author Response

Comments and Suggestions for Authors

Dear Authors,

Although you made some improvements to your paper, I still consider that some important aspects have not been properly revised.

Response:

Dear Reviewer,

Thank you very much for taking the time to review this manuscript. Please find the detailed responses below and the corresponding revisions in track changes in the re-submitted file.

Point-by-point response to Comments and Suggestions for Authors

Comments 1:

[Introduction

Lines 71-76 are difficult to follow and do not seem to have any connection o the purpose of your study. Moreover, since you conduct a study regarding prevalence and other characteristics of OLK (gender distribution, histological subtypes etc.), the appropriate treatment strategy is not the object of your research.]

Response 1: Thank you for pointing this out!  In order to make this paragraph more coherent and in connection with the present study, we have rephrased lines 73-78 as follows:

Our research supports the continuous debate in the scientific literature regarding the onset of oral squamous cell carcinoma (OSCC), particularly in the absence of identified risk factors[17-19]. The development of OSCC may be related to a number of genetic factors or different epigenetic subtypes and mutations [20,21], or a compromised immune system [22]. Research on oral leukoplakia in individuals with not known risk factors may provide insight in the study of OSCC development.” 

Comments 2: 

Material and Methods:

In this section, it is mentioned that this was a retrospective cohort study. If so, how could the patients offer their informed consent? For these types of study, an ethics committee generally gives the approval for the reasonable use of available clinical and histological data.  Again, the treatment protocol is not relevant for your research and since it is guided by a reference published in 2015, how could it be applied to cases identified before 2015?

Response 2

Thank you for your question. All our patients signed an informed consent which mentions the possible inclusion in future studies conducted in the clinical department. We also obtained approval from the Ethical Committee of the University regarding the present research ethics which is mentioned in the study.  We removed from the manuscript the guidelines for OLK treatment and rephrased the sentences as follows:

Lines 106-188:”The treatment selection was individualized for each patient, local irritative factors were removed  and smoking reduction or cessation was recommended.”

Comments 3: 

Results:

This section is still difficult to follow, you mention many percentages and results of Chi-square-tests in the text, which is actually appropriate for a table. I still consider that Table 2 should have included also the descriptive data for the male patients. Just because they did not reach statistical significance, it does not mean they are not worth being mentioned. Apart from that, it should be kept in mind, that on the other side, the pure achievement of statistical significance does not automatically imply a relevant clinical significance.Since the follow- up time of the patients was so variable, ranging from 1 to 228 months, it is difficult to make clear assumptions regarding malignant transformation, as patients who were longer followed have higher odds of being at one point diagnosed with a malignant transformation. Moreover, you included the ex-smokers in the smokers group, which I find reasonable, however, it still matters how long ago these quit smoking.  

Response 3

Thank you for your suggestions! In Table 2 we added a section only for male group with all the results. The malignant transformation rate is the result of our research and we agree this data  reflects the follow up period which we intended to be as long as possible, but we cannot control it due to patient addressability.  

For ex-smokers, we included patients who have quit smoking for at least 1 year. Lines 124-126  “The cohort was divided into two groups: 108 smokers (including 23 ex-smokers at the time of diagnosis - patients who have quit smoking for at least 1 year) and 46 non-smokers”. 

Comments 4:

Discussion and Conclusions:

In spite of the previously mentioned concerns, the Conclusion section has not been improved. It should have been clearly mentioned, that your conclusion is more a clinical observation, maybe randomly encountered, and not a statistically and clinically validated statement.

Response 4

Thank you for your suggestions! We improved the Conclusions as follows in lines 294-302:

Our study revealed statistical differences between OLK in smokers and non-smokers concerning gender- a higher proportion of females in the non-smokers group. The most commonly affected areas were smokers' buccal mucosa and non-smokers' gums. The homogeneous type was the most encountered clinical form of OLK in all patients and was unrelated to smoking behavior. The presence of dysplasia did not significantly differ between smokers and non-smokers.

We report that the non-smokers had an almost twofold higher rate of malignant transformation than the smokers, and that all of the non-smokers' malignant transformation cases were in women.”

Comments 5: Comments on the Quality of English Language

The English language has not been improved, the topic of phrases is still difficult to follow. 

Response 5:

Thank you for your recommendation! We revisited the whole manuscript and made changes marked by track changes.

Reviewer 4 Report

Comments and Suggestions for Authors

Dear Authors,

 Thank you for submitting the revised manuscript. All the suggested changes have been thoroughly addressed, and I have no further comments. I recommend the manuscript for acceptance.

Best regards,

Author Response

Dear Reviewer,

Thank you for taking the time to review our manuscript. Thank you for your comments!

The Authors